# A novel emergency medical services protocol to improve treatment time for large vessel occlusion strokes

Nancy Glober[1]*, Michael Supples[1], Sarah Persaud[1], David Kim[2], Mark Liao[1], Michele Glidden[1], Dan O'Donnell[1], Christopher Tainter[3], Malaz Boustani[1], Andreia Alexander[1]

1 Department of Emergency Medicine, Indiana University School of Medicine, Indianapolis, Indiana, United States of America, 2 Department of Emergency Medicine, Stanford University, Santa Clara County, California, United States of America, 3 Department of Anesthesiology Critical Care, University of California at San Diego, San Diego, California, United States of America

* nglober@iu.edu

**Data Availability Statement:** All relevant data are within the manuscript and its Supporting information files.

## Abstract

In many systems, patients with large vessel occlusion (LVO) strokes experience delays in transport to thrombectomy-capable centers. This pilot study examined use of a novel emergency medical services (EMS) protocol to expedite transfer of patients with LVOs to a comprehensive stroke center (CSC). From October 1, 2020 to February 22, 2021, Indianapolis EMS piloted a protocol, in which paramedics, after transporting a patient with a possible stroke remained at the patient's bedside until released by the emergency department or neurology physician. In patients with possible LVO, EMS providers remained at the bedside until the clinical assessment and CT angiography (CTA) were complete. If indicated, the paramedics at bedside transferred the patient, via the same ambulance, to a nearby thrombectomy-capable CSC with which an automatic transfer agreement had been arranged. This five-month mixed methods study included case-control assessment of use of the protocol, number of transfers, safety during transport, and time saved in transfer compared to emergent transfers via conventional interfacility transfer agencies. In qualitative analysis EMS providers, and ED physicians and neurologists at both sending and receiving institutions, completed e-mail surveys on the process, and offered suggestions for process improvement. Responses were coded with an inductive content analysis approach. The protocol was used 42 times during the study period; four patients were found to have LVOs and were transferred to the CSC. There were no adverse events. Median time from decision-to-transfer to arrival at the CSC was 27.5 minutes (IQR 24.5–29.0), compared to 314.5 minutes (IQR 204.0–459.3) for acute non-stroke transfers during the same period. Major themes of provider impressions included: incomplete awareness of the protocol, smooth process, challenges when a stroke alert was activated after EMS left the hospital, greater involvement of EMS in patient care, and comments on communication and efficiency. This pilot study demonstrated the feasibility, safety, and efficiency of a novel approach to expedite endovascular therapy for patients with LVOs.

**Funding:** NG - K12 HS026390 Agency for Healthcare Research and Quality https://www.ahrq.gov/funding/index.html sponsors did not play any role in the study design, data collection and analysis, decision to publish, or preparation of the manuscript.

**Competing interests:** The authors have declared that no competing interests exist.

## Introduction

In the United States, approximately 795,000 people sustain a stroke each year, which equates to one stroke every 40 seconds, and a death from stroke every four minutes.[1] In 2015, strokes accounted for $66 billion in costs, projected to increase to $143 billion by 2035 [2]. 87% of strokes are acute ischemic strokes (AIS) [1], and 24–38% of AIS are large vessel occlusions (LVOs) [3,4], which have greater morbidity and mortality than non-LVO ischemic strokes, but are amenable to endovascular intervention [4–7].

Since 2015, the advent of endovascular thrombectomy (EVT) has significantly improved outcomes in patients with LVOs [5]. The five initial randomized control trials investigating EVT demonstrated a number needed to treat of 2.6 to reduce a patient's 90-day modified Rankin score (mRS) by 1 point, compared to conventional thrombolysis [5]. Minimizing time to EVT is critical to achieving these superior outcomes. 91% of LVO patients achieved functional independence at 90 days (mRS = 0–2) if EVT was performed within 150 minutes of symptom onset, but the probability of functional independence decreased by 10% over the next hour, and by 20% over each subsequent hour [8]. Every 30-minute increase in time-to-EVT reduced the probability of functional independence by 8.3% [9].

Most hospitals are not capable of performing EVT. Estimates using ambulance response times and demographic data suggest that 81% of the US population has potential access to thrombolysis within one hour, and that 56% could access EVT in the same timeframe [10]. Despite this potential, use of EVT remains low, with barriers including delayed recognition of symptoms and presentation to emergency care [10], and delays in hospital transfer that prevent a majority of patients with LVO from receiving EVT and its associated benefits [11–13].

Emergency medical services (EMS) providers face the challenge of identifying patients with stroke in general and LVO in particular, and transporting these patients to the most appropriate facilities. Patients with non-LVO strokes benefit from rapid transport to the nearest thrombolysis-capable acute stroke ready certified hospital, most of which cannot perform EVT [14]. Most EMS systems either transport all patients with suspected stroke to the nearest PSC, requiring patients with LVO to be transferred to an EVT-capable facility, or use a prehospital stroke severity score to identify patients for preferential transport to a Comprehensive Stroke Center (CSC) that can perform EVT. Direct transport to a CSC improves time-to-thrombectomy and functional outcomes for LVO [13,15], but only 26–51% of patients identified by prehospital scores actually have LVOs [16,17], resulting in inappropriate transport of non-LVO patients to sometimes overwhelmed CSCs, and transport of patients with LVOs to PSCs unable to provide definitive treatment (i.e., EVT).

With the exception of extremely rural settings, the 9-1-1 system throughout the United States is designed to deliver an ambulance within 8 minutes. The average 9-1-1 response time in Marion County is 7 minutes. By contrast, the interfacility transfer system, which would typically transport a patient with LVO from a PSC to CSC for EVT, is less widely and immediately available than the 9-1-1 emergency response system. Interfacility transfer systems employ providers trained in critical care medicine who are prepared to respond to the deterioration of complex patients during transport, but ambulances for interfacility transfer cannot generally be procured as rapidly as those for 9-1-1 response. Many patients requiring inter-hospital transport are sufficiently complex to warrant a delay in transfer in exchange for the benefit of more highly trained providers during transport. Patients with LVO are an exception, in which the importance of expedient transfer to an EVT-capable center often supersedes the method of transport.

Here we report on a pilot study of a novel protocol designed to expedite transfer and EVT for patients with LVO via 9-1-1 response ambulances.

## Methods

For five months (October 2020 to February 2021), Indianapolis EMS piloted a novel "Standby-for-Transfer" protocol for patients with suspected stroke. Under this protocol, Indianapolis EMS facilitated the rapid transfer of patients with LVO to an EVT-capable CSC, instead of the conventional process of enlisting critical care transport through the transfer center. Under this protocol, paramedics who transported patients with suspected stroke to the Sidney and Lois Eskenazi Hospital ED stayed at the patient's bedside until either dismissed by the ED physician (if no LVO was found on CT angiography) or directed to transport the patient to nearby Indiana University-Methodist Hospital for EVT (if LVO was found). This was a mixed-methods study. We report quantitative outcomes (uses of the protocol, transfer times compared to non-protocol transfers, adverse events in transfer), as well as qualitative analysis of provider impressions of the new protocol, using an inductive content analysis approach [18–20].

We included under the protocol all patients 18 years or older transported by Indianapolis EMS to Eskenazi Hospital for suspected stroke between October 1, 2020 and February 22, 2021. Eskenazi and Methodist Hospitals are large urban academic hospitals in downtown Indianapolis, located 2.0 miles apart. Patients with possible strokes taken to Eskenazi do not go directly to the CT scanner, but CT scans for those patients are completed and read by radiology at a higher priority. The respective EDs each provide care to more than 100,000 patients per year. Surrounding Marion County has a population of 903,393, which is 52% female, 11% Hispanic, 63% White, 27% Black, 2% Asian and 8% mixed or other races [21]. Median per capita income is $28,566 [21]. Indianapolis EMS is the largest ambulance service in the state of Indiana and the predominant 9-1-1 response and transport agency in Marion County, responding to approximately 120,000 EMS calls per year. Most responses are at the Advanced Life Support level; ambulances are staffed with one paramedic and one emergency medical technician.

Under the pilot protocol, Indianapolis EMS paramedics remained at the bedside in the ED after transporting any patient with a dispatch code of "stroke" or a primary provider impression of "stroke," "altered mental status," or "weakness," until dismissed by the ED physician or bedside neurologist, or directed to transfer the patient to Methodist Hospital for EVT (Fig 1). Inclusion criteria were selected based on historical analysis to optimize sensitivity for stroke detection. In patients with possible LVO, EMS providers remained at bedside through the neurological assessment and CT angiography. If thrombolysis was indicated, an alteplase bolus was given and the infusion started. Prior to protocol implementation, EMS providers were trained on pump management and indications to discontinue thrombolysis infusion (e.g., motor vehicle collision or other trauma). If an LVO was detected via CT angiography, the patient was returned to the same EMS ambulance with the same crew for expedited transfer to Methodist Hospital, with which an auto-accept agreement was previously arranged for LVO EVT candidates. Care was provided by the EMS crews; additional nursing or physician support were not sent with the patient as part of the protocol. The ED physician called the transfer center to communicate patient information, but not to arrange transfer itself.

Throughout the five-month pilot, we queried EMS and hospital electronic medical records and noted patient demographics, instances of protocol use and associated transfers, adverse events during transport, and transfer times. For patients who were transferred, we recorded symptoms, last known well time, National Institutes of Health Stroke Score (NIHSS), time of thrombolytic administration, transfer time, and the patient's final diagnosis.

To estimate protocol-related improvements in transfer times, we compared protocol LVO transfers with other emergent critical care transfers to Methodist Hospital arranged via the conventional transfer process over the same period. The control patients were transferred

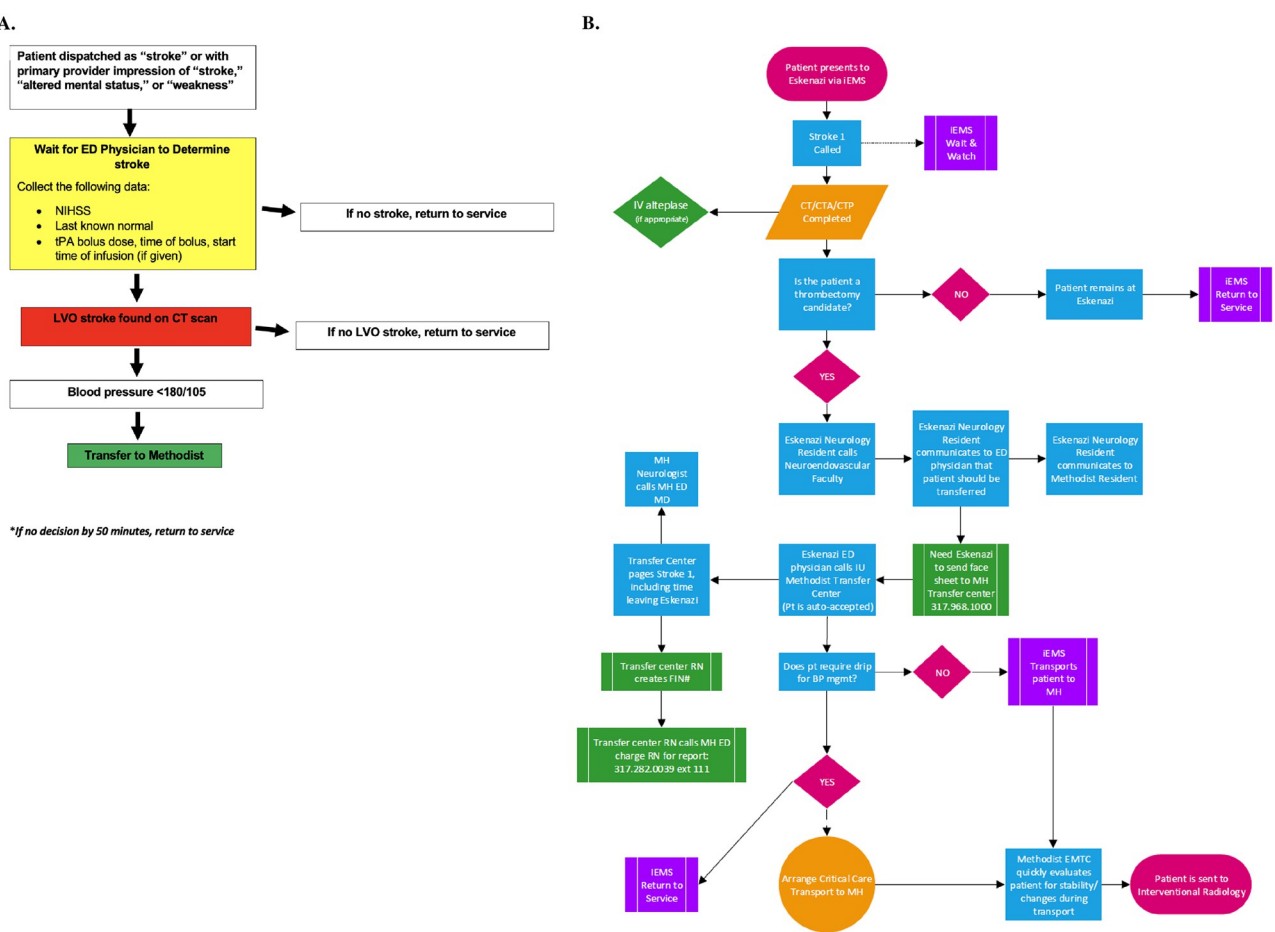

**Fig 1. Novel protocol flowsheets. A)** Brief protocol for Indianapolis EMS providers and **B)** Detailed in-hospital flowsheet.

emergently with the same level of urgency as patients with LVOs. We surveyed EMS providers, and ED physicians and neurologists at both sending and receiving institutions via e-mail. Respondents were asked to respond to the following about the pilot protocol:

1. Tell me about how this process went for you.

2. Tell me how we can improve this process.

3. Tell me how this process compares to what you're used to.

We described the demographics, past medical history, and clinical characteristics of eligible patients. We calculated the time in minutes from the ED physician's decision to transfer an LVO patient to their arrival at Methodist Hospital (median and IQR), and corresponding times for emergent non-protocol transfers to the same facility. We compared transfer times for the two groups with a Wilcoxon rank-sum test using Prism GraphPad (San Diego, California).

Qualitative data were analyzed using Microsoft Word and an inductive content analysis approach [22,23]. Three physicians (one female ED attending and EMS medical director, one male EMS fellow, and one female ED resident) read and openly coded each response as they were completed. All three surveyors had an established relationship with the providers through their professional roles. All providers (EMS and in-hospital) were alerted to the objectives of

the study via electronic-mail prior to the start of the study. The three researchers came together to discuss their individual codes and identify preliminary concepts and themes. Concepts identified in each response were tested in subsequent surveys. The iterative process of coding, recoding, and sub-coding continued until all responses were analyzed and the major themes were identified. Given the design of this mixed-methods study, we did not continue data collection until we reached theoretical saturation. Instead, our goal was to analyze all the responses obtained, which we did. However, by the end of qualitative data analysis we believe we reached theoretical saturation as no new concepts were being identified in the data [24]. Disagreement between researchers was resolved through discussion. Detailed accounts of this coding process were documented in the field journal as part of the audit trail [25].

We reviewed prehospital and in-hospital charts of all patients transferred under the pilot protocol for adverse events, including patient deterioration secondary to transfer by non-critical care trained EMS providers, motor vehicle collisions, and intracranial hemorrhage.

This study was approved by the Indiana University Institutional Review Board (protocol #2008585011).

## Results

### Quantitative analysis

During the study period, 42 patients were brought to Eskenazi Hospital by Indianapolis EMS and evaluated for stroke. Under the "Standby-for-Transfer" protocol, Indianapolis EMS transferred 4 patients with LVO to Methodist Hospital for EVT. Time from patient arrival to administration of tPA did not change significantly when compared to patients in the same time period in the prior year (Table 1).

Patient characteristics are described in Table 2 with further detail of the patients who were transferred in Table 3. Review of patient charts revealed no adverse events to patients or providers associated with transfers under the pilot protocol.

The median time from the Eskenazi ED physician's decision to transfer to patient arrival at Methodist Hospital was 27.5 minutes (IQR 24.5–29.0). During the same time period, 38 patients were emergently transferred from Eskenazi to Methodist Hospital for reasons other than LVO, with a median time from decision-to-transfer to arrival of 314.5 minutes (IQR 204.0–459.3) (data with times and diagnoses included in S1). The difference in transfer times between the two groups was statistically significant at p<0.001. Non-LVO transfers from Eskenazi to Methodist Hospital were most commonly due to a need for a particular specialty or facility (e.g., the cardiac catheterization laboratory at Eskenazi Hospital was unavailable for emergent angiography, or an operating room was unavailable for emergent surgery).

**Table 1. Descriptive statistics of patients receiving alteplase during a period one year prior to the protocol (Oct 1, 2019 to Feb 28, 2020) and during the protocol (Oct 1, 2020 to Feb 28, 2021).**

|  | Pre-Protocol | During Protocol | p-value |
|---|---|---|---|
| Number of Patients | 18 | 14 |  |
| NIHSS—Median (IQR) | 5 (3–12) | 17 (11–26) | <0.001 |
| Door to Needle minutes Median (IQR) | 52 (50–67) | 46 (41–60) | 0.325 |
| Arrival Method |  |  | 0.265 |
| POV | 8 (44.4%) | 9 (64.3%) |  |
| EMS | 10 (55.6%) | 5 (55.75%) |  |

**Table 2. Characteristics of patients transferred during the protocol.**

| | |
|---|---|
| Age—mean (SD) | 60.6 (12.7) |
| **Sex** | |
| Male | 22/42 (52.4%) |
| Female | 20/42 (46.6%) |
| **Race and Ethnicity** | |
| Asian or Pacific Islander | 3/42 (7.2%) |
| Black or African American | 23/42 (54.8%) |
| Hispanic or Latino | 2/42 (4.8%) |
| White | 13/42 (31.0%) |
| **Past Medical History** | |
| Atrial Fibrillation | 9/42 (22.5%) |
| Diabetes | 21/42 (51.2%) |
| Hypertension | 31/42 (75.6%) |
| Hyperlipidemia | 18/42 (45.0%) |
| Prior Stroke | 19/42 (48.7%) |
| Tobacco Use | 29/42 (74.4%) |
| Initial NIHSS (median, IQR) | 7.0 (3.0–13.8) |
| Hours since last known normal (median, IQR) | 2.3 (1.0–5.3) |
| Acute Ischemic Stroke (including LVO) | 15/42 (35.7%) |
| Large Vessel Occlusion | 4/42 (9.5%) |

Demographics, medical history and clinical presentations of patients in the study.

## Qualitative analysis

We sent surveys to 66 consecutive providers involved in the care of study patients, including EMS providers, ED physicians, neurologists, and nurses. EMS providers were surveyed every time the protocol was activated. Physicians and nurses were surveyed when a patient was transferred via EMS. No repeat interviews were carried out. We received responses from 29 EMS providers, 11 ED physicians (3 residents and 8 attendings), 7 neurologists (3 residents

**Table 3. Characteristics of patients transferred using the protocol.**

| Gender | Age | Paramedic RACE Scale | Symptoms | LKW | Time to Arrival at Eskenazi ED | NIHSS | Time from ED Arrival to tPA | Location of LVO | Door-in-door-out time[*] | Disposition |
|---|---|---|---|---|---|---|---|---|---|---|
| Male | 72 | 6 | Right-sided weakness, facial droop and aphasia | 03:30 | 310 min | 17 | Not given | Left MCA at M1/M2 junction | 54 min | Acute rehabilitation facility |
| Male | 47 | 5 | Left sided weakness | 16:00 | 56 min | 17 | 46 min | Right M2 | 76 min | Long-term care facility |
| Male | 33 | undocumented | Right hemiparesis and aphasia | 13:00 | 105 min | 17 | 40 min | Left M2 | 50 min | Discharged home |
| Female | 55 | 6 | Syncope, gaze deviation, right sided weakness | 11:45 | 31 min | 22 | Not given | Left ICA terminus | 54 min | Died in the hospital |

[*]Transfer time was the time the patient left the Eskenazi ED; *LKW*, last known well (time); *NIHSS*, NIH stroke scale; *tPA*, tissue plasminogen activator; *LVO*, large vessel occlusion; *MCA*, middle cerebral artery; *ICA*, internal carotid artery.

and 4 attendings), 1 neurosurgeon/interventionalist, and 3 nurses. Participants did not provide feedback on the findings.

Qualitative analysis of provider responses demonstrated six themes: lack of awareness of a new protocol, communication, smooth process, impediment to using the protocol when a stroke alert was activated after EMS left the hospital, involvement of EMS in patient care, and efficiency. Each theme is discussed in detail below.

**Lack of awareness.** Lack of awareness of a new protocol was a common response from both paramedics and physicians, especially in the first week of protocol use (4 of 11 survey responses). Responses from earlier EMS runs include paramedics saying "I forgot about the protocol" or "I didn't know ['Standby-for-Transfer'] was live." One EMS crew member mentioned that when a decision was made to transfer a patient, "no one seemed to know what to do." One emergency medicine physician at Methodist Hospital remarked "[the] process seemed to work because I didn't know there was a process." Similarly, one paramedic commented that the Methodist ED seemed unaware that a stroke patient was a transferred patient, rather than coming directly from a scene. Overall, numerous responses suggested that not all team members (including paramedics, physicians, and nurses) at either Eskenazi or Methodist Hospitals were aware of the protocol. Later in the study, EMS providers were more likely to be aware of the protocol, but continued to perceive in-hospital providers as lacking awareness of the protocol.

**Communication.** Communication was a common theme mentioned by providers. While several paramedics noted good communication, specifically from physicians to the paramedics, others highlighted a lack of communication between the different team members. Specifically, one paramedic complained, "no one really kept us informed" as the Eskenazi ED providers were assessing a patient for possible transfer. Another paramedic reported, "I reminded everyone in the room of the process" and went on to state they felt the process went smoothly. One ED physician recommended "encourage the EMS crew to check in with the emergency department physician before dispersing," on a case where a stroke was activated after the EMS crew had already left.

**Smooth process.** Ten different paramedics and additional physicians commented that the overall process was "smooth." One paramedic, caring for a patient who did not require transfer, commented "the process was very smooth and we were only there [at Eskenazi] for 15 minutes." An ED physician at Eskenazi who was caring for an LVO patient who was transferred using the protocol described the process as "seamless," and said they "almost didn't notice it happening." Three neurologists provided similar assessments.

**Impediment to using the protocol when a stroke alert was activated after EMS left the hospital.** This theme was raised several times early in the pilot. In one case, an ED resident noted that they decided to activate a stroke alert only after obtaining collateral information from the patient's family, by which point Indianapolis EMS had already left. In another case, an ED resident mentioned that "language barriers and misinformation" led to delayed activation of a stroke alert, which occurred after paramedics had already departed. This theme was also discussed by one emergency medicine attending and one paramedic, possibly for the same patient case(s). This theme was not seen in responses after the first week of the protocol.

**Involvement of EMS in patient care.** Numerous EMS providers made positive comments regarding their increased involvement in patient care when the protocol was used. One paramedic commented, "it was a very educational and fun experience." Another paramedic found it rewarding to witness patient outcomes and stated that the process gave them "closure." Several paramedics found it educational to observe patient care after ED arrival, including discussions over neuroimaging. Overall, EMS crew members made positive remarks regarding their increased involvement in patient care.

**Efficiency.** Many comments touched on efficiency and inefficiency. Several paramedics and in-hospital providers noted inefficiencies related to the protocol, including extended EMS wait time, delays in transporting patients to definitive care, and delays in transfer center communication. One paramedic commented:

"We were out of service for an hour missing several incidents in our area for a patient that likely wasn't even eligible for Methodist's advanced capabilities. If it took 50 minutes to determine if this patient needed intervention, what difference does the 5–10 minutes it would take for an ambulance to respond to Eskenazi make? Not to mention, the hospital would have a difficult time starting all of the interventions needed in that interval and contacting Methodist to confirm approval of transport in a 10-minute time span. . .it's a waste of time and resources."

This sentiment was echoed by other paramedics, as well as by an ED attending physician.

Conversely, other paramedics and physicians frequently cited protocol efficiency. One ED attending responded "it was nice that the patient got to definitive care faster." A neurology resident at Eskenazi also commented on the ease of having paramedics available at bedside to transfer the patient, rather than arranging and waiting for a separate crew, stating "ultimately, the patient likely received a time-sensitive treatment more quickly than they would have if it weren't for EMS being on standby." Furthermore, an emergency medicine resident physician at Methodist Hospital stated the process was "similar if not more efficient than normal stroke alerts. . .because neurology knew about it even before I did." A neurology resident at Methodist commented that the interventional radiologists were "totally prepared for the patient on arrival."

## Discussion

To our knowledge, this is the first study using the United States 9-1-1 response system to expedite transfer of LVO patients for EVT. While this is a pilot study, our data suggest that this protocol can be used to successfully improve time-to-EVT. For EMS systems, using 9-1-1 vehicles for emergent interfacility transfer is unusual, as the priority of the 9-1-1 system is to respond as rapidly as possible to out-of-hospital emergencies. In the case of LVO transfers, however, the relatively small number of patients requiring emergent transfer to EVT-capable centers (as compared to total daily EMS call volume), and the large benefits of earlier EVT for these patients, warranted investigation of such an approach.

While EMS systems vary widely, we believe this approach can be adapted to improve time-to-EVT and corresponding neurological outcomes for patients with LVO. Many studies have demonstrated delays to EVT that are exacerbated when patients initially present to a facility that is not EVT-capable [11–13]. We were interested to note in our data that the use of the protocol led to improvements in time-to-tPA, beyond the improved expediency of utilizing the 9-1-1 ambulance for transfer. Further studies are needed to clarify what types of transfer systems are used nationally, and to what extent delays in transfer are attributable to the limited availability of conventional interfacility transport. While more complex patients (e.g., on mechanical ventilation or requiring multiple infusions) may require a higher provider level of training, this pilot study demonstrates the feasibility of training paramedics to transport LVO patients with ongoing thrombolytic infusions.

EMS systems considering implementation of this protocol must consider potential impact on 9-1-1 response times if an ambulance is out of service for "Standby-for-Transfer." That impact will likely vary widely from system to system, but examination of a similar approach in

Ireland showed similar positive results without adverse impact to their ambulance system [26]. In this study, we did not transfer any patients with intracranial hemorrhage via the pilot protocol. Although a similar approach could be considered in other time-sensitive conditions such as intracranial hemorrhage, trauma and ST segment myocardial infarction, potential benefits of expedited transfer must be weighed against the burden of keeping EMS crews in the ED.

Given the nature of this pilot study, we anticipated some initial period of adjustment. Our qualitative survey results demonstrated a perceived lack of awareness and communication among both in-hospital and prehospital providers. Unsurprisingly, that lack of awareness disappeared in responses over time. We anticipate that communication between in-hospital and prehospital providers would continue to improve with an extended implementation of the protocol. We were pleased that a number of providers commented on the "smoothness" of the novel approach.

One goal of the protocol was to enhance the sense of patient ownership among paramedics, as well as providing education in stroke evaluation, and feedback on patient outcomes. Survey responses indicated that at least some prehospital providers appreciated these opportunities. While it is often impractical to take providers out of service for extended periods, our study shows that there may be specific instances in which longer periods of EMS involvement in the ED can benefit both patients and prehospital providers.

Our protocol did not account for patients who self-presented to the ED, or who were already hospitalized when the need for transfer was identified. In one instance, the protocol was not used because a stroke alert was activated on an ED patient after EMS had already left the hospital.

To improve efficiency in systems with less paramedic capacity, EDs might engage the 9-1-1 system to transfer patients with LVOs, without requiring paramedics to wait at bedside. Through collaborative engagement of EMS medical directors, prehospital providers, neurologists, and ED physicians, we hope to continue to improve our systemwide response with the goal of significantly improving outcomes for patients with LVO by safely reducing time-to-EVT.

## Limitations

This study was limited by the small number of patients for whom the protocol was activated. As a pilot study, the protocol was only in effect for five months. While we were able to assess feasibility, improvements in transfer time, and impact on the EMS system, we were unable to directly assess impact on neurological outcomes. Because we conducted the pilot study during a period when the sending facility (Eskenazi Hospital) was temporarily without thrombectomy capability due to renovations, we cannot directly compare transfer times for LVO patients before and during the study period, since LVO patients prior to the study period could receive thrombectomy at Eskenazi Hospital. However, we used as our control patients who were emergently transferred from Eskenazi to Methodist during the same time period via the conventional process. Finally, while we believe this study provides compelling evidence in favor of using the 9-1-1 system to expedite the transfer of patients requiring emergent thrombectomy, we acknowledge that EMS systems vary widely across the country, and that our approach will require modification to match the resources and practices of any system in which it would be implemented.

## Conclusion

A "Standby-for-Transfer" protocol, using the 9-1-1 emergency response system to expedite transfer of LVO patients to a comprehensive stroke center for endovascular thrombectomy,

demonstrated feasibility and improved transport times, and has the potential to improve patient outcomes for an extremely time-sensitive critical condition.

## Supporting information

**S1 Table. Emergent transfers using the conventional transfer system.**
(XLSX)

## Author Contributions

**Conceptualization:** Nancy Glober, Mark Liao, Michele Glidden, Dan O'Donnell, Christopher Tainter, Malaz Boustani, Andreia Alexander.

**Data curation:** Nancy Glober, Michael Supples, Sarah Persaud, Mark Liao, Michele Glidden.

**Formal analysis:** Nancy Glober, Michael Supples, Sarah Persaud, David Kim, Andreia Alexander.

**Funding acquisition:** Nancy Glober.

**Project administration:** Nancy Glober, Michele Glidden, Dan O'Donnell.

**Supervision:** Nancy Glober, Mark Liao, Michele Glidden, Dan O'Donnell, Andreia Alexander.

**Validation:** Nancy Glober.

**Writing – original draft:** Nancy Glober, Sarah Persaud.

**Writing – review & editing:** Nancy Glober, Michael Supples, David Kim, Michele Glidden, Dan O'Donnell, Christopher Tainter, Malaz Boustani, Andreia Alexander.

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
