## [Decision Letter · Decision Letter 0]

13 Aug 2021

PONE-D-21-24078

A Novel Emergency Medical Services Protocol to Improve Treatment Time for Large Vessel Occlusion Strokes

PLOS ONE

Dear Dr. Glober,

Thank you for submitting your manuscript to PLOS ONE. After careful consideration, we feel that it has merit but does not fully meet PLOS ONE’s publication criteria as it currently stands. Therefore, we invite you to submit a revised version of the manuscript that addresses the points raised during the review process.

Please respond to the reviewers comments below. In particular, some concerns about the technical soundness, analysis methodology and statistical analysis of the data presented have been raised that must be addressed clearly. 

We look forward to receiving your revised manuscript.

Kind regards,

Karen M Doyle, PhD

Academic Editor

PLOS ONE

Journal Requirements:

3. We note you have included a table to which you do not refer in the text of your manuscript. Please ensure that you refer to Table 2 in your text; if accepted, production will need this reference to link the reader to the Table.

Additional Editor Comments:

Please clearly address all issues raised regarding methodology and data analysis.

Please give a complete account of qualitative data analysis approach, such as the methods for research triangulation or coding systems used to select significant sections from participant statements and the process of theme derivation. Information on software used in this analysis should be provided. Statistical analysis of quantitative parameters should be included in as much as possible.

Reviewers' comments:

Reviewer's Responses to Questions

**Comments to the Author**

1. Is the manuscript technically sound, and do the data support the conclusions?

Reviewer #1: Partly

Reviewer #2: Yes

2. Has the statistical analysis been performed appropriately and rigorously? 

Reviewer #1: No

Reviewer #2: Yes

3. Have the authors made all data underlying the findings in their manuscript fully available?

Reviewer #1: Yes

Reviewer #2: Yes

4. Is the manuscript presented in an intelligible fashion and written in standard English?

Reviewer #1: Yes

Reviewer #2: Yes

5. Review Comments to the Author

Reviewer #1: The study of Glober et al is focus on an emergency medical services (EMS) protocol to expedite transfer of patients with LVOs to a comprehensive stroke center (CSC). In wich paramedics, after transporting a patient to non-CSC with a possible stroke remained at the patient’s bedside until released by the emergency department or neurology physician.. If indicated, the paramedics at bedside transferred the patient, via the same ambulance, to a nearby thrombectomy capable. They concluded that “Standby-for-Transfer” protocol demonstrated feasibility and improved transport times.

I agree with the authors that time is brain and improving times in all of the steps is crucial in acute stroke management. Otherwise, there is lack of information in different parts of the protocol that may clarify attention to suspected stroke patients.

I have some major and minor concerns:

The statistical analysis is poor of data and results are based on comparison on groups with different underlying acute medical conditions. I think that including more details in your results would improve the quality of your research .

-Please provide information about the internal protocol of your institution concerning how transfer of a suspected stroke is managed . Are there different steps in emergency room or is the patient moved Directly to perform a Computed Tomography? , Include the reference of Noreen Kamal et al. 2017

-Because this is a study conducted to analyze the influence of paramedicals activity in work flow, I would like to see paramedical scores of suspected strokes for identifying patients like “stroke-like symptoms (termed FAST-positive) ,The RACE Scale ( see work of Robert L Dickson et al . Prehosp Emerg Care. 2019 .) or any other score that are often used by paramedics for screening strokes in your community

-Were there any case of Intracranial hemorraghe stroke ( ICH) and did these patients benefit of your protocol?

-Please include door-in-door-out (DIDO) times of each group , and reference the work from “Ambulance waiting and associated work flow improvement strategies of Eva Gaynor”

-Did your protocol may help to improve times of intravenous fibrinolysis with or without LVO ? Can you provide data of door-to-needle time in those eligible for thrombolysis and can you compare these patients with a historical control group ?

The authors compare patients with suspected strokes with patients with other conditions, please include in the discussion how your protocol may benefit main emergencies and not only suspected LVO strokes .

Reviewer #2: This manuscript reports on outcomes of a 5 month long pilot study using a novel emergency medical services protocol to improve treatment time for large artery occlusive stroke. EMS paramedics EMS providers remained at the bedside until the clinical and imaging assessment of a suspected stroke patient were complete. If indicated, the paramedics at bedside transferred the patient, via the same ambulance, to a nearby thrombectomy capable comprehensive stroke centre (CSC) with which an automatic transfer agreement had been arranged. On quantitative analysis the study demonstrated a significant reduction in median time from decision-to-transfer to arrival at CSC compared with other acute non-stroke emergent interhospital transfers during same time period.

A major limitation of the study is the inability to directly compare the decision-to-transfer times for LVO patients during the pilot study to those prior to the study. Transfer times for other acute non-stroke emergent interhospital transfers during same time period was instead used as a comparison. Presumably these other acute non-stroke emergent interhospital transfers would be undertaken with the same level of urgency as transfer of acute stroke patients – this needs to be clarified and clearly stated, otherwise the comparison is futile.

A further limitation is lack of comparison of other time metrics such as door to needle and door to decision time in stroke patients prior to and during the pilot study.

Were patients transferred with accompanying nursing staff/physicians, or solely the EMS personnel? This requires clarification.

The qualitative analysis of the pilot study is interesting, with useful organization of findings into major themes. This section of the results would however benefit from being shortened.

6. PLOS authors have the option to publish the peer review history of their article (what does this mean?). If published, this will include your full peer review and any attached files.

Reviewer #1: No

Reviewer #2: No

---

## [Author Response · Author response to Decision Letter 0]

15 Sep 2021

Thank you so much for the thoughtful comments from reviewers. We are pleased to resubmit our article A Novel Emergency Medical Services Protocol to Improve Treatment Time for Large Vessel Occlusion Strokes with the following responses:

Please ensure that your manuscript meets PLOS ONE's style requirements.

We reviewed the style requirements and revised accordingly.

Upon re-submitting your revised manuscript, please upload your study’s minimal underlying data set as either Supporting Information files or to a stable, public repository 

We included a supplemental file with our data. 

We note you have included a table to which you do not refer in the text of your manuscript. Please ensure that you refer to Table 2 in your text; if accepted, production will need this reference to link the reader to the Table.

We edited to include reference to Table 2 in our text.

Please give a complete account of qualitative data analysis approach, such as the methods for research triangulation or coding systems used to select significant sections from participant statements and the process of theme derivation. Information on software used in this analysis should be provided. Statistical analysis of quantitative parameters should be included in as much as possible.

We included details of the software used and statistical analysis used on the quantitative data.

The statistical analysis is poor of data and results are based on comparison on groups with different underlying acute medical conditions. I think that including more details in your results would improve the quality of your research.

We listed the limited number of patients in our analysis as a limitation in the manuscript. While it does limit our conclusions, we hope that the presentation of a novel approach and our findings presented here garner interest in the manuscript. We have included more detail on the comparison group and their different medical problems. Specifically, we included in S1 the transfer time for each patient used in the control group and also the medical reason for emergent transfer.

Please provide information about the internal protocol of your institution concerning how transfer of a suspected stroke is managed. Are there different steps in emergency room or is the patient moved Directly to perform a Computed Tomography? 

We detailed the protocol in the emergency room.

Because this is a study conducted to analyze the influence of paramedicals activity in work flow, I would like to see paramedical scores of suspected strokes for identifying patients like “stroke-like symptoms (termed FAST-positive) ,The RACE Scale ( see work of Robert L Dickson et al . Prehosp Emerg Care. 2019 .) or any other score that are often used by paramedics for screening strokes in your community

We included the RACE scale by paramedics of each patient in table 2.

Were there any case of Intracranial hemorraghe stroke ( ICH) and did these patients benefit of your protocol?

There were no cases of intracranial hemorrhage stroke transferred via this protocol. We added commentary in the discussion section.

Please include door-in-door-out (DIDO) times of each group, and reference the work from “Ambulance waiting and associated work flow improvement strategies of Eva Gaynor”

We included DIDO times for the transferred patients add added the suggested reference.

Did your protocol may help to improve times of intravenous fibrinolysis with or without LVO ? Can you provide data of door-to-needle time in those eligible for thrombolysis and can you compare these patients with a historical control group?

We included further data on time to intravenous fibrinolysis during the protocol and before the protocol.

The authors compare patients with suspected strokes with patients with other conditions, please include in the discussion how your protocol may benefit main emergencies and not only suspected LVO strokes.

We included commentary in the discussion section.

A major limitation of the study is the inability to directly compare the decision-to-transfer times for LVO patients during the pilot study to those prior to the study. Transfer times for other acute non-stroke emergent interhospital transfers during same time period was instead used as a comparison. Presumably these other acute non-stroke emergent interhospital transfers would be undertaken with the same level of urgency as transfer of acute stroke patients – this needs to be clarified and clearly stated, otherwise the comparison is futile.

We included further detail of the type of control patients and clearly stat that the control patients were transferred with the same level of urgency.

A further limitation is lack of comparison of other time metrics such as door to needle and door to decision time in stroke patients prior to and during the pilot study.

We added data on door to needle time prior to and during the pilot study. Because of the nature of the study (occurring during an Eskenazi thrombectomy suite remodel) we are not able to provide door to decision time outside of the study.

Were patients transferred with accompanying nursing staff/physicians, or solely the EMS personnel? This requires clarification.

The patients were transferred with EMS personnel. We clarified in the text.

The qualitative analysis of the pilot study is interesting, with useful organization of findings into major themes. This section of the results would however benefit from being shortened.

We shortened the qualitative analysis section.

---

## [Decision Letter · Decision Letter 1]

13 Oct 2021

PONE-D-21-24078R1A Novel Emergency Medical Services Protocol to Improve Treatment Time for Large Vessel Occlusion StrokesPLOS ONE

Dear Dr. Glober,

Thank you for submitting your manuscript to PLOS ONE. After careful consideration, we feel that it has merit but does not fully meet PLOS ONE’s publication criteria as it currently stands. Therefore, we invite you to submit a revised version of the manuscript that addresses the points raised during the review process.

Reviewer 1 has voiced concerns about the robustness of the data and statistical analysis. These concerns must be satisfactorily addressed in order for this manuscript to be accepted for publication in PlosOne.

We look forward to receiving your revised manuscript.

Kind regards,

Karen M Doyle, PhD

Academic Editor

PLOS ONE

Reviewers' comments:

Reviewer's Responses to Questions

**Comments to the Author**

1. If the authors have adequately addressed your comments raised in a previous round of review and you feel that this manuscript is now acceptable for publication, you may indicate that here to bypass the “Comments to the Author” section, enter your conflict of interest statement in the “Confidential to Editor” section, and submit your "Accept" recommendation.

Reviewer #1: (No Response)

Reviewer #2: All comments have been addressed

2. Is the manuscript technically sound, and do the data support the conclusions?

Reviewer #1: No

Reviewer #2: (No Response)

3. Has the statistical analysis been performed appropriately and rigorously? 

Reviewer #1: No

Reviewer #2: (No Response)

4. Have the authors made all data underlying the findings in their manuscript fully available?

Reviewer #1: No

Reviewer #2: (No Response)

5. Is the manuscript presented in an intelligible fashion and written in standard English?

Reviewer #1: Yes

Reviewer #2: (No Response)

6. Review Comments to the Author

Reviewer #1: The authors made some efforts to improve the quality of the manuscript, but there is still lack of data and robust statistical analysis.

As explained by the author “ Time from arrival via EMS to administration of tPA improved from median time of 59.8 minutes when the protocol was not in use to 43 minutes during the protocol (p<0.01).”

I cannot see were this affirmation comes from. This data should be demonstrated through a table of case-control analysis . Because the analysis of modified transfer times was conducted during 5 months, I would suggest to compare this data with a hystorical control group of at least 5 months in the same period of the last year .

It is of outmost importance to incorporate number of patients treated with iv fibrinolysis as well as door-to-needle time (DNT) to iv fibrinolysis between both groups.

“Table 1. Patient Characteristics” should be changed to a comparison analysis of historical control group vs analyzed group.

The most important part of the manuscript is about reduction of times of treatment based on your protocol, in page 12 “(data with times and diagnoses included in S1) I guess that this means supplementary material S1 but I cannot find any table or figure in S1 including these KEY data.

So I ask again about my two previous questions to be answered

“Please include door-in-door-out (DIDO) times of each group, and reference the work from “Ambulance waiting and associated work flow improvement strategies of Eva Gaynor”

We included DIDO times for the transferred patients add added the suggested reference.

WHERE IS THIS DATA?

Did your protocol may help to improve times of intravenous fibrinolysis with or without LVO ? Can you provide data of door-to-needle time in those eligible for thrombolysis and can you compare these patients with a historical control group?

We included further data on time to intravenous fibrinolysis during the protocol and before the protocol.

WHERE IS THIS DATA?

On the other hand the part of results that includes literally comments in pages 13 to 17 “ “ , I suggest to be included in a supplementary material because it does not improve the quality of the manuscript.

The aim of the study should be to incorpotate the protocol “Standby-for-Transfer” to improve times of treatment for iv fibrinolysis and transfer times of LVO strokes, so the results and the discussion should be related to this purpose.

Reviewer #2: All comments or recommendations from this reviewer have been addressed by the authors of the manuscript.

7. PLOS authors have the option to publish the peer review history of their article (what does this mean?). If published, this will include your full peer review and any attached files.

Reviewer #1: No

Reviewer #2: No

---

## [Author Response · Author response to Decision Letter 1]

14 Dec 2021

Thank you for the reviewers’ comments. We are pleased to resubmit our article A Novel Emergency Medical Services Protocol to Improve Treatment Time for Large Vessel Occlusion Strokes with the responses listed below. 

The authors made some efforts to improve the quality of the manuscript, but there is still lack of data and robust statistical analysis. As explained by the author “ Time from arrival via EMS to administration of tPA improved from median time of 59.8 minutes when the protocol was not in use to 43 minutes during the protocol (p<0.01).” I cannot see were this affirmation comes from. This data should be demonstrated through a table of case-control analysis. Because the analysis of modified transfer times was conducted during 5 months, I would suggest to compare this data with a hystorical control group of at least 5 months in the same period of the last year.

It is of outmost importance to incorporate number of patients treated with iv fibrinolysis as well as door-to-needle time (DNT) to iv fibrinolysis between both groups. “Table 1. Patient Characteristics” should be changed to a comparison analysis of historical control group vs analyzed group.

We added a table 1 to show this for period from Oct-Feb 2019/2020 compared to Oct-Feb 2020/2021. When we used a historical control (as opposed to 5 months pre protocol and 5 months during protocol). 

The most important part of the manuscript is about reduction of times of treatment based on your protocol, in page 12 “(data with times and diagnoses included in S1) I guess that this means supplementary material S1 but I cannot find any table or figure in S1 including these KEY data.

We included Table S1 at the end of the body of the manuscript (as opposed to a separate upload) to hopefully facilitate reviewer’s ability to see the data. 

So I ask again about my two previous questions to be answered

“Please include door-in-door-out (DIDO) times of each group, and reference the work from “Ambulance waiting and associated work flow improvement strategies of Eva Gaynor” We included DIDO times for the transferred patients add added the suggested reference.

WHERE IS THIS DATA?

The reference to Eva Gaynor is in paragraph 3 of the discussion section, sentence 2. 

Column 10 of Table 3 outlines the DIDO times for the stroke patients who were transferred. If the reviewer is referring to DIDO times for control patients, as discussed in the manuscript this data does not exist. The pilot study occurred in a time period when the EVT suite for the sending facility was being remodeled. In the year prior, there were no DIDO times because the patients were not being transferred. Similarly, DIDO times are not commonly collected for non-stroke emergent transfers as this is not a common metric in other pathologies. 

Did your protocol may help to improve times of intravenous fibrinolysis with or without LVO ? Can you provide data of door-to-needle time in those eligible for thrombolysis and can you compare these patients with a historical control group? We included further data on time to intravenous fibrinolysis during the protocol and before the protocol.

WHERE IS THIS DATA?

As mentioned above, we added a table 1 to show this for period from Oct-Feb 2019/2020 compared to Oct-Feb 2020/2021. The time to intravenous fibrinolysis is additionally described in paragraph 1 of the results section.

On the other hand the part of results that includes literally comments in pages 13 to 17 “ “ , I suggest to be included in a supplementary material because it does not improve the quality of the manuscript.

We respectfully submit that a mixed methods approach is legitimate and highlights the important provider reactions to the pilot of a radically different protocol from that commonly used. For other groups attempting to pursue or operationalize a similar protocol in their system, understanding cultural context and reactions is important, not supplemental.

The aim of the study should be to incorporate the protocol “Standby-for-Transfer” to improve times of treatment for iv fibrinolysis and transfer times of LVO strokes, so the results and the discussion should be related to this purpose.

We did not find that the protocol improved times to fibrinolysis but we attempted to further emphasize the focus on improved time-to-EVT in the discussion section. We hope that by adding the new Table 1 in the results section we adequately addressed this concern regarding the results.

---

## [Decision Letter · Decision Letter 2]

14 Feb 2022

A Novel Emergency Medical Services Protocol to Improve Treatment Time for Large Vessel Occlusion Strokes

PONE-D-21-24078R2

Dear Dr. Glober,

We’re pleased to inform you that your manuscript has been judged scientifically suitable for publication and will be formally accepted for publication once it meets all outstanding technical requirements.

Kind regards,

Karen M Doyle, PhD

Academic Editor

PLOS ONE

Additional Editor Comments (optional):

Reviewers' comments:

Reviewer's Responses to Questions

**Comments to the Author**

1. If the authors have adequately addressed your comments raised in a previous round of review and you feel that this manuscript is now acceptable for publication, you may indicate that here to bypass the “Comments to the Author” section, enter your conflict of interest statement in the “Confidential to Editor” section, and submit your "Accept" recommendation.

Reviewer #1: All comments have been addressed

Reviewer #2: All comments have been addressed

2. Is the manuscript technically sound, and do the data support the conclusions?

Reviewer #1: Yes

Reviewer #2: (No Response)

3. Has the statistical analysis been performed appropriately and rigorously? 

Reviewer #1: Yes

Reviewer #2: (No Response)

4. Have the authors made all data underlying the findings in their manuscript fully available?

Reviewer #1: Yes

Reviewer #2: (No Response)

5. Is the manuscript presented in an intelligible fashion and written in standard English?

Reviewer #1: Yes

Reviewer #2: (No Response)

6. Review Comments to the Author

Reviewer #1: The authors improved the quality of the paper and replied to my concerns satisfactory so I agree to approve the submitted paper to be published .

Reviewer #2: (No Response)

7. PLOS authors have the option to publish the peer review history of their article (what does this mean?). If published, this will include your full peer review and any attached files.

Reviewer #1: No

Reviewer #2: No

---

## [Editor Report · Acceptance letter]

16 Feb 2022

PONE-D-21-24078R2 

A Novel Emergency Medical Services Protocol to Improve Treatment Time for Large Vessel Occlusion Strokes 

Dear Dr. Glober:

I'm pleased to inform you that your manuscript has been deemed suitable for publication in PLOS ONE. Congratulations! Your manuscript is now with our production department. 

Kind regards, 

on behalf of

Dr. Karen M Doyle 

Academic Editor

PLOS ONE